# Alternating Reinforcement Learning for Rubric-Based Reward Modeling in Non-Verifiable LLM Post-Training

Ran Xu [* 1]   Tianci Liu [* 2]   Zihan Dong [3]   Tony Yu [4]   Ilgee Hong [4]
Carl Yang [1]   Linjun Zhang [3]   Tuo Zhao [4]   Haoyu Wang [5]

## Abstract

Standard reward models typically predict scalar scores that fail to capture the multifaceted nature of response quality in non-verifiable domains, such as creative writing or open-ended instruction following. To address this limitation, we propose RUBRIC-ARM, a framework that jointly optimizes a rubric generator and a judge using reinforcement learning from preference feedback. Unlike existing methods that rely on static rubrics or disjoint training pipelines, our approach treats rubric generation as a latent action learned to maximize judgment accuracy. We introduce an alternating optimization strategy to mitigate the non-stationarity of simultaneous updates, providing theoretical analysis that demonstrates how this schedule reduces gradient variance during training. Extensive experiments show that RUBRIC-ARM achieves strong performance among baselines on multiple benchmarks and significantly improves downstream policy alignment in both offline and online reinforcement learning settings.

## 1. Introduction

Reward modeling serves as the compass for aligning large language models (LLMs) with human intents, typically by generating a scalar score or preference label to predict human preferences (Stiennon et al., 2020; Wang et al., 2024a). However, in complex non-verifiable domain, such as creative writing or open-ended instruction following, these scalar or pairwise judgments often fail to capture the multifaceted nature of response quality (Ying et al., 2025). To

address this limitation, recent advancements have shifted toward rubric-based reward modeling, where models explicitly generate structured criteria to ground their judgments (Gunjal et al., 2025; Liu et al., 2025a; Pathak et al., 2025). By decomposing evaluation into interpretable dimensions, rubric-based models offer transparency and improve generalization across prompt-specific evaluation axes.

Central to rubric-based evaluation is the availability of *high-quality rubrics*. To ensure rubric quality, earlier work has primarily relied on human-authored rubrics, which are expensive to produce and difficult to scale to large datasets (Arora et al., 2025). More recent approaches seek to automate rubric construction using LLMs (Viswanathan et al., 2026; Gunjal et al., 2025); however, these methods are largely prompting-based and rely on fixed, frozen models for both rubric generation and response quality judgment. Consequently, they do not update the model's intrinsic capabilities to the target domain or the underlying preference distribution, limiting their ability to generate in-domain, preference-aligned rubrics. Moreover, even when learning-based components are introduced (Liu et al., 2025a; Rezaei et al., 2025), the rubric generator and the judge are treated as separate modules and trained independently rather than jointly optimized. This decoupled training pipeline prevents deeper integration between rubric construction and judgment, leading to suboptimal evaluation signals. Designing effective rubric-based reward models are still challenging.

In this work, we propose RUBRIC-ARM, an end-to-end framework that jointly optimizes the *rubric generator* and the *judge* via alternating reinforcement learning (RL), enabling the two components to co-evolve and mutually reinforce one another during training. We formulate rubrics as *latent actions* that guide the reward model in recovering the underlying preference signal, and posit that improved rubric generation directly leads to more accurate preference predictions. To ensure stable joint optimization, RUBRIC-ARM employs an alternating training strategy that decouples the learning dynamics while preserving a shared objective. Training alternates between (i) optimizing the reward model with a fixed rubric generator to align with target preference labels, and (ii) optimizing the rubric generator with

---

[*]These authors contributed equally to this work; author order was determined by randomization through a die roll. [1]Emory University [2]Purdue University [3]Rutgers University [4]Georgia Institute of Technology [5]University at Albany. Correspondence to: Ran Xu <ran.xu@emory.edu>, Tianci Liu <liu3351@purdue.edu>, Haoyu Wang <hwang28@albany.edu>.

*Proceedings of the 43$^{rd}$ International Conference on Machine Learning*, Seoul, South Korea. PMLR 306, 2026. Copyright 2026 by the author(s).

a fixed reward model to produce discriminative rubrics that maximize prediction accuracy.

A key challenge of the alternating RL is the instability caused by simultaneous updates to both components. Our analysis reveals that early-stage exploration by the rubric generator can dominate the learning dynamics. To mitigate this, we first stabilize the reward model under fixed rubrics before optimizing the rubric generator. This alternating schedule reduces variance and ensures robust optimization.

Our contributions can be summarized as follows:

- We develop RUBRIC-ARM, a rubric-based reward model to produce high-quality rubrics and precise judgments. To the best of our knowledge, this is the first approach that jointly optimizes rubric and judging via RL.

- We introduce an *alternating RL* training algorithm that couples the rubric generator and judge through a shared correctness objective, enabling mutual improvement while stabilizing optimization.

- We evaluate RUBRIC-ARM across diverse alignment settings (9 reward modeling and 6 policy benchmarks). RUBRIC-ARM outperforms strong reasoning-based judges and prior rubric-based reward models, achieving a $+4.7\%$ average gain on reward-modeling benchmarks, and consistently improves downstream policy post-training when used as the reward signal.

**Conflict of Interest Disclosure.** The authors declare no financial conflicts of interest related to this work. The evaluation includes several third-party models and API-based systems, but none of the authors has a financial relationship with the organizations developing these systems that could reasonably be perceived as affecting the work.

## 2. Related Works

**LLM-based Reward and Judge Models.** While Zheng et al. (2023) established the foundational utility of LLM-based judges. Subsequent research expanded the scope of reasoning to include chain-of-thoughts (Zhang et al., 2025b), self-critiques (Ankner et al., 2024; Yu et al., 2025b; Mahan et al., 2024) or plan evaluations strategically (Saha et al., 2025). Liu et al. (2025c) explore inference-time reasoning for generative reward models. Recent studies (Chen et al., 2025a;b; Whitehouse et al., 2025; Guo et al., 2026; Hong et al., 2025; Xu et al., 2026) leverage online RL to directly incentivize detailed reasoning, aiming to mitigate bias and enhance the accuracy of pointwise and pairwise scoring.

**Rubrics-based Reward Models.** Recently, rubric-based approaches have emerged as a promising direction for LLM evaluation (Arora et al., 2025; Hashemi et al., 2024; Pathak et al., 2025; Akyürek et al., 2025), alignment (Viswanathan

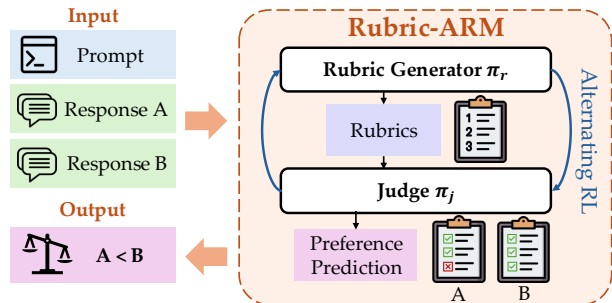

*Figure 1.* The overall framework for RUBRIC-ARM.

et al., 2026; Zhang et al., 2025a; Wang et al., 2025a), and reasoning (Gunjal et al., 2025; Zhou et al., 2025; Huang et al., 2025). However, a unique challenge lies in generating *high-quality rubrics at scale*. To address this, Li et al. (2026); Liu et al. (2025a); Xie et al. (2026) extract rubrics from pairwise comparison signals, while (Rezaei et al., 2025; Zhang et al., 2025a; Shao et al., 2025) dynamically generate rubrics by leveraging policy model outputs in an online setting.

## 3. Preliminaries

We study rubric-based reward modeling in *non-verifiable* domains, where response quality cannot be directly validated against ground truth. The rubric-based reward model contains two parts, namely *rubric generator* and *judge*. The key components of RUBRIC-ARM are described as follows.

**Rubrics.** We define a rubric as a structured set of evaluation criteria conditioned on a prompt. Formally, let $x$ denote a prompt, a rubric $r(x) = \{c_i\}_{i=1}^{k}$ consists of $k$ criteria, where each $c_i$ specifies a distinct aspect of response quality (e.g., factual correctness, tone, or presentation).

For training rubric-based reward models in non-verifiable domains, a pairwise preference dataset is given as $\mathcal{D} = \{(x_i, y_i^{(1)}, y_i^{(2)}, o_i^\star)\}_{i=1}^{N}$, where $x$ is a prompt, $y^{(1)}$ and $y^{(2)}$ are two candidate responses, and $o^\star \in \{0, 1\}$ indicates which response is preferred (e.g., $o^\star = 1$ means $y^{(1)} \succ y^{(2)}$). Formally, the **rubric generator** $\pi_r$ generates a rubric $r$ from the prompt as

$$r \sim \pi_r(\cdot \mid x; \theta_r), \tag{1}$$

while a **judge** $\pi_j$ predicts a preference $o$ with the reasoning chain $c$ conditioned on the prompt, responses, and rubric as

$$(c, o) \sim \pi_j(\cdot \mid x, y^{(1)}, y^{(2)}, r; \theta_j). \tag{2}$$

**Objective.** We define the preference-correctness reward

$$R(o, o^\star) = \mathbb{I}[o = o^\star], \tag{3}$$

where $\mathbb{I}[o = o^\star]$ represents if the binary prediction extracted from $o$ aligns with ground truth $o^\star$.

Denote $\theta_r, \theta_j$ as the parameter for $\pi_r$ and $\pi_j$ respectively, our goal is to learn $(\theta_r, \theta_j)$ that maximize expected prefer-

ence correctness under generated rubrics:

$$\max_{\theta_r,\theta_j} \mathbb{E}_{(x,y^{(1)},y^{(2)},o^\star)\sim\mathcal{D}} \mathbb{E}_{r\sim\pi_r(\cdot|x;\theta_r)}$$
$$\mathbb{E}_{(c,o)\sim\pi_j(\cdot|x,y^{(1)},y^{(2)},r;\theta_j)} \left[ R(o,o^\star) \right]. \tag{4}$$

Since both $r$ (text) and $c$, $o$ (discrete decision with reasoning) are sampled actions, we optimize Equation (4) with RL.

## 4. RUBRIC-ARM: Alternating RL for Rubric Generation and Judging

In non-verifiable domains, supervision is limited to pairwise preference feedback and rubrics are not directly observed. Simultaneously updating the rubric generator $\pi_r$ and the judge $\pi_j$ leads to non-stationary learning targets and unstable optimization. As shown in Figure 1, RUBRIC-ARM addresses this challenge using an alternating RL scheme that decouples the updates of two components.

### 4.1. Stage I: SFT Warmup

We equip both $\pi_j$ and $\pi_r$ with basic rubric generation and judging capabilities via leveraging open-source datasets. Following the prior work (Liu et al., 2025a), we fine-tune on synthetic rubrics and judge trajectories derived from open-source datasets including *UltraFeedback* (Cui et al., 2024), *SkyWork* (Liu et al., 2024), *Magpie* (Xu et al., 2025b), and *Synthetic Instruction Following* (Lambert et al., 2025a). Both $\pi_r(r \mid x;\theta_r)$ and $\pi_j(c,o \mid x,y^{(1)},y^{(2)},r;\theta_j)$ are trained with the standard next-token prediction objective.

### 4.2. Stage II: Alternating Reinforcement Learning

Stage I (SFT) warm-starts the rubric generator $\pi_r$ and judge $\pi_j$ by imitating synthetic rubric generation and judging trajectories, but optimizes the two components independently and does not directly target preference correctness. We therefore optimize both components using *alternating reinforcement learning*. Specifically, training switches between (i) *improving the judge with a fixed rubric generator* and (ii) *improving the rubric generator with a fixed judge*, providing each component with a clearer learning signal while preserving the same end objective $R(o,o^\star)$.

**(i) RL for Judge $\pi_j$ with the current $\pi_r$.** With the rubric generator parameters $\theta_r$ held fixed, we update $\theta_j$ to improve preference correctness under rubrics sampled from $\pi_r$:

$$\max_{\theta_j} J_j(\theta_j;\theta_r) = \mathbb{E}_{(x,y^{(1)},y^{(2)},o^\star)\sim\mathcal{D}} \mathbb{E}_{r\sim\pi_r(\cdot|x;\theta_r)}$$
$$\mathbb{E}_{(c,o)\sim\pi_j(\cdot|x,y^{(1)},y^{(2)},r;\theta_j)} \left[ \mathbb{I}[o = o^\star] \right]. \tag{5}$$

This phase trains the judge to produce rubric-conditioned evaluations that recover the dataset preference.

Since $\pi_r(\cdot \mid x;\theta_r)$ is fixed during judge updates, we cache rubrics to reduce sampling cost and stabilize optimization.

For each training instance $(x_i,y_i^{(1)},y_i^{(2)},o_i^\star)$, we sample a rubric $r_i \sim \pi_r(\cdot \mid x_i;\theta_r)$ once and reuse it for multiple judge optimization steps, yielding the Monte Carlo estimate:

$$J_j(\theta_j;\theta_r) \approx \mathbb{E}_{(x_i,y_i^{(1)},y_i^{(2)},o_i^\star)\sim\mathcal{D},\, r_i}$$
$$\mathbb{E}_{(c,o)\sim\pi_j(\cdot|x_i,y_i^{(1)},y_i^{(2)},r_i;\theta_j)} \left[ \mathbb{I}[o = o_i^\star] \right]. \tag{6}$$

In practice, we use a shaped reward that augments the final correctness signal $R_{\text{acc}} = \mathbb{I}[o = o_i^\star]$ with *format-based* reward $R_{\text{fmt}}$ that enforces valid judging trajectories (i.e., addressing each rubric criterion with per-criterion explanations, followed by an overall justification and a final decision). The final reward for the judge $\pi_j$ is $R_j = R_{\text{acc}} + R_{\text{fmt}}$.

**(ii) RL for Rubric Generator $\pi_r$ with the current $\pi_j$.** With the judge parameters $\theta_j$ fixed, we update $\theta_r$ to prefer rubrics that lead the current judge to make correct decisions. Concretely, we maximize the preference correctness under rubrics drawn from $\pi_r$ as:

$$\max_{\theta_r} J_r(\theta_r;\theta_j) = \mathbb{E}_{(x,y^{(1)},y^{(2)},o^\star)\sim\mathcal{D}} \mathbb{E}_{r\sim\pi_r(\cdot|x;\theta_r)}$$
$$\mathbb{E}_{(c,o)\sim\pi_j(\cdot|x,y^{(1)},y^{(2)},r;\theta_j)} \left[ \mathbb{I}[o = o^\star] \right]. \tag{7}$$

Intuitively, $\pi_r$ learns to generate criteria that are discriminative for the given prompt and usable by the judge to recover the dataset preference. In practice, we approximate the expectation with a single rollout by greedy decoding ($t = 0$), i.e., we generate one judging trajectory $(c,o)$ per rubric and use the Monte Carlo estimate

$$R_r = \mathbb{I}[o = o^\star]. \tag{8}$$

**Optimization (alternating RL).** RUBRIC-ARM alternates between optimizing Eq. 5 and 7. At iteration $t$, we run:

$$r_i^t \sim \pi_r(\cdot \mid x_i;\theta_r^t) \quad \forall (x_i,y_i^{(1)},y_i^{(2)},o_i^\star) \in \mathcal{D}, \tag{9}$$
$$\theta_j^{t+1} \leftarrow \text{GRPO}\left(\theta_j^t;\{r_i^t\},\mathcal{D}\right), \tag{10}$$
$$\theta_r^{t+1} \leftarrow \text{GRPO}\left(\theta_r^t;\theta_j^{t+1},\mathcal{D}\right). \tag{11}$$

Here we cache one rubric per instance during judge updates (since $\pi_r$ is fixed in that phase). In each phase, GRPO (Shao et al. (2024), details in Appendix A) updates only the active policy while keeping the other fixed. Notably, we alternate training by updating the judge before the rubric generator in each cycle. In Sec. 5, we provide theoretical analysis proving the benefits of this ordering.

### 4.3. Policy Model Post-training with RUBRIC-ARM

We use the trained rubric generator $\pi_r(\cdot \mid q;\theta_r)$ and judge $\pi_j(\cdot \mid q,\cdot,\cdot,r;\theta_j)$ to provide preference supervision for post-training a policy model $\pi_\phi(a \mid q)$, where $q$ denotes the prompt and $a$ denotes a candidate response. For any pair of responses $(a,b)$, RUBRIC-ARM samples a rubric $r \sim \pi_r(\cdot \mid q;\theta_r)$ and predicts a preference label

$$\widehat{o} = \text{Judge}_{\theta_j}(q,a,b,r) \in \{0,1\}, \tag{12}$$

where $\widehat{o} = 0$ indicates $a \succ b$ and $\widehat{o} = 1$ indicates $b \succ a$.

**Preference Optimization with RUBRIC-ARM.** Given a prompt $q$, we sample two rollouts from the current policy,

$$a_1, a_2 \sim \pi_\phi(\cdot \mid q), \qquad (13)$$

and use RUBRIC-ARM to label which one is preferred via Eq. (12) and retain examples where the predictions are consistent for both orders. We then update $\pi_\phi$ with the standard DPO objective (Rafailov et al., 2023) relative to a fixed reference policy $\pi_{\text{ref}}$. For iterative DPO (Xiong et al., 2024; Pang et al., 2024), we repeat (i) sampling rollouts, (ii) labeling them with RUBRIC-ARM, and (iii) applying DPO updates for multiple rounds.

**Online RL with RUBRIC-ARM.** Following recent works on using pairwise judges to provide reward signals (Xu et al., 2025a), we also consider online RL where RUBRIC-ARM provides rewards for optimizing $\pi_\phi$. For each prompt $q$, we adopt the ReMax-style baseline construction (Li et al., 2024) by first generating a deterministic reference response via greedy decoding,

$$a^{(0)} = \text{Greedy}(\pi_\phi(\cdot \mid q)) \quad (t = 0), \qquad (14)$$

and then sample $K$ additional rollouts,

$$\{a^{(k)}\}_{k=1}^K \sim \pi_\phi(\cdot \mid q). \qquad (15)$$

To mitigate positional bias, we query the judge in both orders under the same rubric $r$. Let $\widehat{o}_\rightarrow^{(k)} \in \{0, 1\}$ denote the judge outcome for $(q, a^{(k)}, a^{(0)}, r)$ and $\widehat{o}_\leftarrow^{(k)} \in \{0, 1\}$ for the swapped order $(q, a^{(0)}, a^{(k)}, r)$. We define the final reward for response $a^{(k)}$ as

$$R_\phi(q, a^{(k)}) = \frac{1}{2}\left(\mathbb{I}(\widehat{o}_\rightarrow^{(k)} = 0) + \mathbb{I}(\widehat{o}_\leftarrow^{(k)} = 1)\right). \qquad (16)$$

## 5. Theoretical Analysis

We analyze the gradient variance to justify our training schedule. We compare two phases: **Strategy A** (Judge Warmup), where we optimize the judge with pre-generated, reused rubrics; and **Strategy B** (Rubric Generator Training), where we optimize the rubric generator against a fixed judge.

**Setup.** Let $u_r(r) := \nabla \log \pi_r(r \mid x)$ and $u_j(o \mid r) := \nabla \log \pi_j(o \mid c, r)$ be the score functions. Let $p(r) := \mathbb{P}(o = o^* \mid c, r)$ be the judge's correctness probability given a rubric. We define the gradient variance as $\text{Var}(\widehat{g}) := \mathbb{E}\|\widehat{g}\|^2 - \|\mathbb{E}[\widehat{g}]\|^2$.

### 5.1. Variance Decomposition

We first examine Strategy A. By freezing the rubric $\bar{r}$ (reuse) during judge updates, we eliminate inter-rubric variance.

**Proposition 5.1** (Judge Variance under Strategy A). *Conditioned on a reused rubric $\bar{r}$, the variance of the judge's*

gradient estimator $\widehat{g}_A$ is solely determined by the judge's binary classification uncertainty:

$$\text{Var}(\widehat{g}_A \mid \bar{r}) = p(\bar{r})(1 - p(\bar{r}))\|u_j(o^* \mid \bar{r})\|^2. \qquad (17)$$

**Proposition 5.2** (Generator Variance under Strategy B). *The total variance of the generator's gradient estimator $\widehat{g}_B$ decomposes into:*

$$\text{Var}(\widehat{g}_B) = \underbrace{\mathbb{E}_r\left[p(r)(1 - p(r))\|u_r(r)\|^2\right]}_{\textit{(I) Multiplicative Reward Noise}} + \underbrace{\text{Var}_r(p(r)u_r(r))}_{\textit{(II) Cross-Rubric Inconsistency}} \qquad (18)$$

**Interpretation.** Term (I) represents the judge's Aleatoric uncertainty amplified by the high-dimensional generator gradient $\|u_r\|^2$. Term (II) captures the optimization difficulty when different rubrics yield different expected rewards $p(r)$, causing the gradient direction to oscillate.

### 5.2. Variance Domination in Early Training

We now derive the variance gap. Instead of assuming trivial gradient dominance, we postulate a condition linking the generator's exploration intensity to its gradient magnitude.

**Assumption 5.3** (Exploration-Gradient Sufficiency). We assume that during early training, the generator's gradient norm is sufficient relative to the judge's, satisfying the following exploration-dependent lower bound:

$$\frac{\|u_r\|}{\|u_j\|} > \sqrt{\frac{1 - p(r)}{1 - p(r) + C_1 p(r)}}, \qquad (19)$$

where $p$ represents the judge's correctness probability (analyzed pointwise or in expectation), and $C_1 \in (0, 1)$ is defined as: $C_1 := \text{Var}_r(p(r)u_r(r))/\mathbb{E}_r[p(r)^2\|u_r(r)\|^2]$.

*Remark* 5.4. The condition in Assumption 5.3 is mild and physically justified. Active exploration ($C_1 > 0$) introduces a positive buffer, making the required gradient-norm ratio on the RHS strictly less than 1 and thus avoiding the need for the generator's gradient to strictly dominate. Moreover, the judge and generator both produce comparable-length sequences over the same vocabulary (checks/prediction vs. rubrics), so their gradient norms are typically of the same order; the exploration buffer is enough to absorb small mismatches and satisfy the condition in practice.

**Theorem 5.5** (Strict Variance Domination). *Under Assumption 5.3, the gradient variance of Strategy B strictly dominates the expected conditional variance of Strategy A:*

$$\text{Var}(\widehat{g}_B) > \mathbb{E}_{\bar{r}}[\text{Var}(\widehat{g}_A \mid \bar{r})]. \qquad (20)$$

*This inequality establishes that the **structural instability** driven by exploration (quantified by $C_1$) is the governing factor in the variance landscape, overriding differences in gradient magnitudes.*

*Remark* 5.6 (Implication for Training Stability). The variance gap derived in Theorem 5.5 justifies the proposed train-

*Table 1.* Comparison of different judge and reward models across multiple benchmarks. RewardBench2 reports results on Precise IF, and Focus dimensions. Rubric API uses GPT-4.1-Mini, and Judge API uses Gemini-2.5-Flash-Lite. Best results are highlighted in **bold**.

| | RewardBench | | IF Evaluation Benchmarks | | | | RM-Bench | RewardBench2 | | HelpSteer3 | Avg. |
|---|---|---|---|---|---|---|---|---|---|---|---|
| | Chat | Chat Hard | FollowBench | PPE-IFEval | InfoBench | IFBench | Chat | Precise IF | Focus | | |
| *Black-box LLMs (For reference only)* | | | | | | | | | | | |
| Claude-3.5-Sonnet | 96.4 | 74.0 | – | 58.0 | – | – | 62.5 | 38.8 | 87.0 | – | - |
| Gemini-2.5-Flash | 95.0 | 83.3 | 86.0 | 75.0 | 85.6 | 69.3 | 78.5 | 57.5 | 84.1 | 70.6 | 78.5 |
| API (Rubric+Judge) | 79.6 | 79.2 | 83.2 | 61.0 | 82.2 | 66.2 | 67.9 | 42.5 | 79.6 | 71.4 | 71.3 |
| API (direct Judge) | 89.6 | 71.2 | 81.7 | 59.2 | 72.9 | 60.4 | 67.2 | 13.2 | 63.4 | 70.3 | 64.9 |
| *Larger White-box LLMs (For reference only)* | | | | | | | | | | | |
| RM-R1-14B (Qwen-2.5-Inst) | 73.5 | 79.8 | 84.0 | 59.0 | 85.5 | 60.8 | 73.2 | 23.8 | 84.6 | 74.8 | 69.9 |
| RM-R1-14B (DeepSeek-Dist) | 90.3 | 78.9 | 89.9 | 61.2 | 82.4 | 59.0 | 71.4 | 30.6 | 79.0 | 74.6 | 71.7 |
| RM-R1-32B (Qwen-2.5-Inst) | 95.3 | 80.3 | 84.9 | 60.4 | 86.1 | 60.4 | 75.3 | 33.1 | 84.2 | 72.9 | 73.3 |
| RM-R1-32B (DeepSeek-Dist) | 95.3 | 83.1 | 89.2 | 63.2 | 85.0 | 58.6 | 74.2 | 36.9 | 79.2 | 75.6 | 74.0 |
| RRM-32B | 94.7 | 81.1 | 85.7 | 60.2 | 84.4 | 60.8 | 73.9 | 34.4 | 83.6 | 75.4 | 73.4 |
| *White-box Judge/Reward LLMs* | | | | | | | | | | | |
| RM-R1-7B (Qwen-2.5-Inst) | 83.0 | 70.0 | 56.3 | 55.2 | 71.3 | 55.2 | 64.2 | 20.6 | 76.2 | 65.2 | 61.7 |
| RM-R1-7B (DeepSeek-Dist) | 85.3 | 67.3 | 69.7 | 51.0 | 70.3 | 56.5 | 62.2 | 13.8 | 55.4 | 62.6 | 59.4 |
| RRM-7B | 77.7 | 69.5 | 65.5 | 51.0 | 68.2 | 53.2 | 59.9 | 10.0 | 60.4 | 62.4 | 57.8 |
| JudgeLRM-7B | **92.1** | 56.1 | 79.8 | 46.0 | 62.7 | 47.5 | 55.4 | 9.4 | 29.1 | 60.2 | 53.8 |
| *Rubric-based Methods* | | | | | | | | | | | |
| Qwen-3-8B (Rubric+Judge) | 73.9 | 63.6 | 63.0 | 53.8 | 74.6 | 55.6 | 64.2 | 21.9 | 56.6 | 61.8 | 58.9 |
| RUBRIC-RM | 88.2 | 74.1 | 76.1 | 67.0 | 80.8 | 65.4 | 65.7 | 34.4 | 82.2 | 67.0 | 70.1 |
| RUBRIC-RM-voting@5 | 89.9 | 75.4 | 81.5 | 70.8 | 83.8 | **67.1** | 67.0 | 40.0 | 86.5 | 67.5 | 73.0 |
| RUBRIC-ARM | 89.4 | 79.6 | 85.7 | 70.8 | 86.1 | 65.9 | **69.2** | 41.9 | 89.4 | 69.8 | 74.8 |
| RUBRIC-ARM-voting@5 | 90.3 | **80.7** | **87.4** | **72.0** | **87.7** | **67.1** | 69.1 | **46.2** | **90.3** | **71.1** | **76.2** |

ing schedule (We first train the judge, then train the rubric generator, and subsequently perform alternating training following this sequence.) by highlighting a critical trade-off in Signal-to-Noise Ratio (SNR). The strictly higher variance in Strategy B implies that generator updates are dominated by exploration stochasticity rather than the true gradient direction, risking optimization instability. In contrast, Strategy A acts as a *variance reduction* mechanism: by fixing the rubric, it effectively sets the exploration coefficient $C_1 \to 0$ locally, isolating the judge from structural noise and providing a stable target for effective learning.

# 6. Experiment

## 6.1. Datasets and Experiment Settings

**Training data.** We train the two components of RUBRIC-ARM, the *rubric generator* and the *judge*, on the general-domain portions of OPENRUBRICS (Liu et al., 2025a). The dataset is split equally into non-overlapping parts, and each rubric-judge alternating round is run on a single part. During training judge, we randomly shuffle the order of response candidates to be evaluated; as shown in App. D.2, this practice greatly helps reduce position bias in reward modeling.

**Backbone and variants.** Both the rubric generator and the judge are fine-tuned from Qwen-3-8B (Team, 2025). At inference time, RUBRIC-ARM follows the two-stage rubric-judging process, as detailed in Sec. 3. We also report ensemble results *voting@5*, by aggregating five independent judges via majority voting.

**Baselines.** For reward-model evaluation, we follow Liu et al. (2025a) and compare RUBRIC-ARM against strong same-scale white-box judges, including JudgeLRM (Chen et al., 2025a), RRM (Guo et al., 2026), RM-R1 (Chen et al., 2025b), and RUBRIC-RM (Liu et al., 2025a) (SFT-only rubric generator + judge). We also report judges using black-box APIs when available. To isolate the benefit of rubric-aware training, we include a training-free baseline, QWEN-3-8B (RUBRIC+JUDGE) (Yang et al., 2025), which directly generates rubrics and judgments via prompting. For policy training, we use RUBRIC-ARM as the reward model to fine-tune Qwen2.5-7B-Instruct (Qwen et al., 2025) and compare against Skywork (Liu et al., 2024), ArmoRM (Wang et al., 2024b), UltraFeedback (Cui et al., 2024), RLCF/AI Judge (Viswanathan et al., 2026), OnlineRubrics (Rezaei et al., 2025), and RUBRIC-RM (Liu et al., 2025a).

**Evaluation benchmarks and metrics.** We evaluate RUBRIC-ARM as a pairwise reward model on widely used alignment benchmarks: RewardBench (Chat/Chat-Hard) (Lambert et al., 2025b), RM-Bench (Liu et al., 2025b), PPE-IFEval (Frick et al., 2024), FollowBench (Jiang et al., 2024), InfoBench (Qin et al., 2024), IFBench (Peng et al., 2025), RewardBench2 (Precise-IF/Focus) (Malik et al., 2025), Arena-Hard (Chiang et al., 2024), AlpacaEval 2 (Dubois et al., 2025), Creative Writing Benchmark v3 (Paech, 2025), WildBench (Lin et al., 2024), and Writing-PreferenceBench (Ying et al., 2025). For FollowBench and InfoBench, we convert the original single-response setup to pairwise evaluation by sampling two responses from the same model (Qwen-3-8B/14B) and using the benchmark's

*Table 2.* Ablation study about the effectiveness of the format reward and the order of judge optimization and rubric generator. Best results are highlighted in **bold**.

| | RewardBench | | IF Evaluation Benchmarks | | | | RM-Bench | RewardBench2 | | HelpSteer3 | Avg. |
|---|---|---|---|---|---|---|---|---|---|---|---|
| | Chat | Chat Hard | FollowBench | PPE-IFEval | InfoBench | IFBench | Chat | Precise IF | Focus | | |
| RUBRIC-ARM switch opt | 93.2 | 76.3 | 85.9 | 67.3 | 84.1 | 64.6 | 69.5 | 24.4 | 86.1 | 71.8 | 72.4 |
| RUBRIC-ARM switch opt-voting@5 | **94.0** | 76.5 | **89.1** | 67.8 | 85.0 | 64.6 | **69.8** | 39.4 | 90.1 | **72.4** | 74.9 |
| RUBRIC-ARM w/o format | 89.8 | 78.7 | 87.1 | 69.2 | 86.1 | 64.3 | 69.5 | 25.6 | 84.8 | 70.8 | 72.6 |
| RUBRIC-ARM w/o format-voting@5 | 91.5 | 78.5 | 88.2 | 70.2 | 87.7 | 65.1 | 69.7 | 43.8 | 88.9 | 71.1 | 75.5 |
| RUBRIC-ARM | 89.4 | 79.6 | 85.7 | 70.8 | 86.1 | 65.9 | 69.2 | 41.9 | 89.4 | 69.8 | 74.8 |
| RUBRIC-ARM-voting@5 | 90.3 | **80.7** | **87.4** | **72.0** | **87.7** | **67.1** | 69.1 | **46.2** | **90.3** | 71.1 | **76.2** |

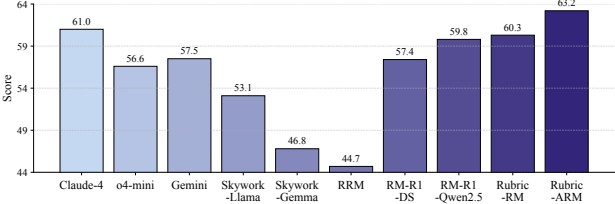

*Figure 2.* Performance of different judge and reward models on WritingPreferenceBench.

verifier to identify constraint violations. We follow each benchmark's official splits and scoring rules, reporting accuracy, win-rate, or the benchmark-specific metric.

### 6.2. Performance of RUBRIC-ARM

Table 1 compares RUBRIC-ARM against a broad set of judge/reward models. RUBRIC-ARM achieves the best average performance among all white-box methods, improving RUBRIC-RM from 70.1 to 74.8, and reaching 76.2 with voting@5. These gains are consistent across both instruction-following and preference-style benchmarks, supporting our key contribution: RUBRIC-ARM learns *more discriminative rubrics* and a *more reliable rubric-conditioned judge* through RL. Notably, RUBRIC-ARM also substantially outperforms API-based judges (e.g., 76.2 vs. 71.3 for Rubric+Judge API and 64.9 for direct Judge API), indicating that explicit rubric-conditioned learning yields a stronger and more stable evaluation signal than black-box judging.

We further assess generalization on WRITINGPREFER-ENCEBENCH (Ying et al., 2025), shown in Fig. 2 (detail results are shown in Table 12), which serves as an out-of-distribution benchmark since none of the compared reward/judge models are trained on this domain. Despite this distribution shift, RUBRIC-ARM remains strong and achieves the best overall score among all methods (63.2), outperforming RUBRIC-RM (60.3) and strong reasoning reward models such as RM-R1-Qwen2.5-7B (59.8). The improvements are broad across diverse writing genres (e.g., Functional, Promotional, Non-Fiction, and Poetry), suggesting that RUBRIC-ARM learns rubrics that capture transferable criteria beyond the training domains, thereby providing a robust reward signal with improved OOD generalization.

*Table 3.* Comparison of trained policy models with different reward models on a format-based constrained instruction-following benchmark (IFEval) and an open-ended benchmark (InfoBench). Baseline results with "⋆" are from Viswanathan et al. (2026); Liu et al. (2025a). Results with underlines are reproduced by us using official checkpoints and evaluation scripts. Best scores are in **bold**.

| Model | IFEval (Prompt) | | IFEval (Inst.) | | IFEval | InfoBench |
|---|---|---|---|---|---|---|
| | Loose | Strict | Loose | Strict | AVG | AVG |
| GPT-4 (0314)⋆ | 79.3 | 76.9 | 85.4 | 83.6 | 81.3 | 87.3 |
| AutoIF (Dong et al., 2025) | 56.9 | 47.1 | 67.0 | 57.6 | 57.2 | 80.6 |
| UltraIF (An et al., 2025) | 75.4 | 71.3 | 83.0 | 79.4 | 77.3 | 80.7 |
| RAIF (Qin et al., 2025) | – | – | – | – | 70.1 | 82.7 |
| Qwen2.5-7B-Instruct⋆ | 75.0 | 72.5 | 81.8 | 79.9 | 77.3 | 78.1 (76.0) |
| + SFT (Distilled)⋆ | 66.8 | 64.1 | 75.3 | 72.8 | 69.8 | 72.5 |
| + DPO (via Skywork)⋆ | 75.7 | 68.0 | 83.2 | 78.5 | 76.0 | 82.0 |
| + DPO (via ArmoRM)⋆ | 73.8 | 70.2 | 81.7 | 78.3 | 76.0 | 83.5 |
| + DPO (via Ultrafbk.)⋆ | 71.5 | 69.1 | 79.9 | 77.7 | 74.6 | 80.0 |
| + DPO (via AI Judge)⋆ | 73.0 | 68.9 | 80.9 | 77.8 | 75.2 | 76.1 |
| + DPO (via RLCF)⋆ | 77.3 | 72.6 | 84.1 | 80.3 | 78.6 | 84.1 (81.5) |
| + IterDPO (via RLCF) | 78.2 | 74.3 | 84.5 | 81.1 | 79.5 | 81.8 |
| + DPO (via RUBRIC-RM)⋆ | 78.2 | 73.9 | 84.5 | 81.2 | 79.5 | 83.0 |
| + IterDPO (via RUBRIC-RM) | 77.6 | 74.1 | 84.3 | 81.7 | 79.4 | 83.3 |
| + DPO (via RUBRIC-ARM) | 78.7 | 76.0 | 84.7 | 82.5 | 80.4 | 83.7 |
| + IterDPO (via RUBRIC-ARM) | **79.3** | 75.1 | **86.0** | **82.9** | 80.8 | **85.0** |

### 6.3. Ablation Study

Table 2 reports two ablation studies that examine (i) the optimization order between the judge and the rubric generator, and (ii) the contribution of the format reward. Unless stated otherwise, all settings are kept identical to RUBRIC-ARM.

**Optimization order.** Our default schedule updates the judge first, then the rubric generator, and alternates thereafter. Swapping this order (`switch opt`) consistently hurts performance: the average drops from $74.8 \to 72.4$ $(-2.4)$ without voting and from $76.2 \to 74.9$ $(-1.3)$ with voting@5, with especially large regressions on strict instruction-following metrics (e.g., RewardBench2-Precise IF: $41.9 \to 24.4$). This suggests that a stronger judge provides a less noisy learning signal for rubric optimization.

**Format reward.** Removing the format reward (`w/o format`) also degrades results: $74.8 \to 72.6$ $(-2.2)$ without voting and $76.2 \to 75.5$ $(-0.7)$ with voting@5. The largest gains appear on structure-sensitive metrics (e.g., RewardBench2-Precise IF: $+16.3$), indicating that $R_{fmt}$ helps prevent degenerate judging behaviors (e.g., missing criteria checks) and improves rubric adherence.

*Table 4.* Comparison of different strategies applied to Qwen2.5-7B-Instruct on **Arena-Hard** and **AlpacaEval**. Results are reported for vanilla models and style/length-controlled settings. Baseline results with "⋆" are from Viswanathan et al. (2026); Rezaei et al. (2025); Liu et al. (2025a). Best results are in **bold**.

| Model | Arena-Hard | | AlpacaEval | | AVG |
|---|---|---|---|---|---|
| | **Vanilla** | **Style-Con** | **Vanilla** | **Length-Con** | |
| GPT-4 (0314)⋆ | 50.0 | 50.0 | 22.1 | 35.3 | 39.4 |
| UltraIF (An et al., 2025) | 31.4 | – | – | – | – |
| Qwen2.5-7B-Instruct⋆ | 51.3 | 42.8 | 33.5 | 36.2 | 41.0 |
| + SFT (Distilled)⋆ | 32.6 | 29.2 | 36.1 | 33.3 | 32.8 |
| + DPO (via Skywork)⋆ | 55.1 | 50.3 | 44.8 | 41.5 | 47.9 |
| + DPO (via ArmoRM)⋆ | 50.8 | 46.4 | 37.6 | 38.1 | 43.2 |
| + DPO (via Ultrafbk.)⋆ | 52.8 | 47.9 | 33.7 | 38.7 | 43.3 |
| + DPO (via AI Judge)⋆ | 51.0 | 44.4 | 28.8 | 33.4 | 39.4 |
| + DPO (via RLCF)⋆ | 54.6 | 48.4 | 36.2 | 37.1 | 44.1 |
| + IterDPO (via RLCF) | 51.1 | 54.6 | 38.9 | 39.2 | 46.0 |
| + DPO (via RUBRIC-RM)⋆ | 52.9 | 53.1 | 47.0 | 41.3 | 48.6 |
| + IterDPO (via RUBRIC-RM) | 56.3 | 56.7 | 50.1 | 42.0 | 51.3 |
| + RL (via ONLINERUBRICS)⋆ | 56.5 | – | **55.0** | 30.4 | – |
| + DPO (via RUBRIC-ARM) | 57.8 | **59.5** | 47.1 | 42.5 | 51.7 |
| + IterDPO (via RUBRIC-ARM) | **58.8** | 58.9 | 52.0 | **44.0** | **53.4** |

*Table 5.* Comparison of different alignment strategies applied to Qwen2.5-7B-Instruct on **WildBench**. Results are reported for task-specific scores and task macro WB score. Baseline results with "⋆" are from Wang et al. (2025b). Best results are in **bold**.

| Method | Creative | Planning | Math | Info seeking | Coding | WB Score |
|---|---|---|---|---|---|---|
| Claude-3.5-Sonnet (20240620)⋆ | 55.6 | 55.6 | 50.2 | 55.5 | 56.5 | 54.7 |
| GPT-4-turbo (20240409)⋆ | 58.7 | 56.2 | 51.0 | 57.2 | 55.1 | 55.2 |
| GPT-4o-mini (20240718)⋆ | 60.1 | 58.2 | 54.0 | 57.4 | 57.2 | 57.1 |
| Qwen2.5-7B-Instruct⋆ | 50.1 | 51.8 | 47.1 | 50.7 | 45.0 | 48.7 |
| +DRIFT⋆ | 52.5 | 53.2 | 50.6 | 52.4 | 50.3 | 51.7 |
| +SPIN⋆ | 43.3 | 45.5 | 41.6 | 46.3 | 39.1 | 42.9 |
| +IterDPO⋆ (via OpenAssistant) | 46.8 | 48.6 | 44.5 | 48.0 | 44.3 | 46.3 |
| +DPO (via RLCF) | 51.4 | 52.7 | 49.0 | 51.3 | 48.8 | 50.5 |
| +IterDPO (via RLCF) | 51.9 | 52.6 | 47.8 | 51.4 | 46.5 | 49.7 |
| +DPO (via RUBRIC-RM) | 54.8 | 55.5 | 51.5 | 54.1 | 52.9 | 53.6 |
| +IterDPO (via RUBRIC-RM) | 57.0 | 56.2 | 50.6 | 54.9 | 52.8 | 54.0 |
| +DPO (via RUBRIC-ARM) | 55.2 | 55.6 | 49.5 | 56.0 | 53.1 | 53.7 |
| +IterDPO (via RUBRIC-ARM) | **57.3** | **57.2** | **53.3** | **56.2** | **55.2** | **55.7** |

## 6.4. Performance of offline RL-based Policy Models

We evaluate whether reward models trained by RUBRIC-ARM transfer to downstream *offline* policy learning.

**Instruction-Following Evaluation.** Table 3 and Fig. 3 show that policies optimized with RUBRIC-ARM-trained rewards consistently achieve the strongest instruction-following performance. On IFEval, DPO with RUBRIC-ARM improves the overall average to 80.4, and iterative DPO further raises it to 80.8 (best), with particularly strong gains on instruction-level constraints. The advantage also transfers to the open-ended InfoBench benchmark, where RUBRIC-ARM reaches 83.7 with DPO and 85.0 with iterative DPO (best). Compared to iterative baselines, RUBRIC-ARM remains consistently stronger: on IFBench (Fig. 3), RLCF improves from 28.2 to 32.0 with IterDPO, while RUBRIC-ARM achieves 35.4 with IterDPO; similarly, iterative RUBRIC-RM reaches 33.7, still below RUBRIC-ARM. Overall, these results indicate that RUBRIC-ARM provides a more precise reward signal, and that iterative optimization amplifies the gains over both one-shot DPO and iterative baselines.

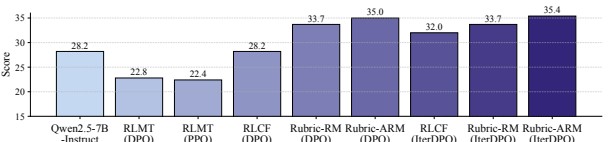

*Figure 3.* Comparison of trained policy models on IFBench. Results of baselines except Rubric-RM (IterDPO) are from Open-Rubrics (Liu et al., 2025a).

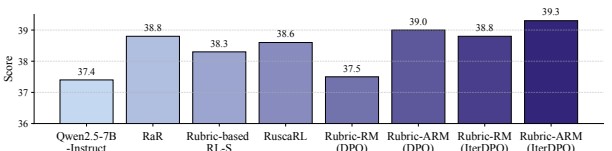

*Figure 4.* Comparison of trained policy models on Create Writing Benchmark v3. Results of baselines except Rubric-RM are from RuscaRL (Zhou et al., 2025).

**Human Preference Alignment Evaluation.** Table 4 and Table 5 show that RUBRIC-ARM-trained rewards consistently yield stronger preference alignment across both controlled and open-domain evaluations. On Arena-Hard and AlpacaEval (Table 4), DPO with RUBRIC-ARM achieves the best overall average (51.7), and IterDPO further improves it to 53.4 (best). On WildBench (Table 5), RUBRIC-ARM again yields the strongest macro score: DPO via RUBRIC-ARM reaches 53.7, while IterDPO via RUBRIC-ARM achieves 55.7 (best), improving over IterDPO with RUBRIC-RM (54.0) by 1.7%, indicating improved preference-aligned helpfulness on broad, real-world tasks.

**Creative Writing.** We further evaluate whether RUBRIC-ARM-based rewards benefit open-ended generation on the Creative Writing Benchmark v3 (Fig. 4). Policies trained with RUBRIC-ARM outperform baselines: DPO using RUBRIC-ARM achieves 39.0, and IterDPO further improves to 39.3 (best). Notably, RUBRIC-ARM-based offline optimization also surpasses strong creative-writing baselines such as RAR (38.8) and RUSCARL (38.6), suggesting that rewards learned by RUBRIC-ARM generalize well to subjective, non-verifiable generation tasks beyond standard instruction following and preference alignment.

## 6.5. Performance of online RL-based Policy Models

We evaluate RUBRIC-ARM in an *online* RL setting by directly optimizing QWEN2.5-7B-INSTRUCT with GRPO (Sec. 4.3) using different reward models. As shown in Table 7, GRPO with RUBRIC-ARM-trained rewards substantially improves both instruction following and preference alignment compared to the base model and a strong reward baseline RM-R1. Specifically, QWEN2.5-7B-INSTRUCT achieves an average score of 46.8, while GRPO with RM-R1 increases it to 52.3. Replacing the reward with RUBRIC-ARM yields the best overall performance, reaching 55.4

*Table 6.* Case study with error highlighting. Baselines either lose in the reasoning process, or favor the longer response but miss the explicit constraint about thumb war, while RUBRIC-ARM enforces hard rules before evaluating principles.

| Case Study on RewardBench Chat Hard | |
| --- | --- |
| **Prompt** | Wars involve armed conflicts between nations, states, groups, or individuals, resulting in the loss of life, property damage, displacement of populations, and numerous other types of suffering. Is a *thumb war* violent? I care about both physical and psychological violence. |
| **Resp A (snippet)** | "War is inherently violent ... direct physical violence ... psychological violence ..." (*"war"; not "thumb war"*) |
| **Resp B (snippet)** | " ... not considered a violent activity but rather a fun and harmless pastime ... friendly competition ..." |
| **Label** | Resp B. |
| **RRM-7B** | " ... Assistant 1 ... doesn't directly answer ... But wait ... the instruction doesn't specify only thumb war ... it's more about war in general ... I'm a bit confused ... (No Decision)" |
| **JudgeLRM** | "Assistant 1 provided a detailed and accurate response ... Assistant 2's answer is inaccurate ... (Choose A)" |
| **RUBRIC-ARM** | |
| **Rubric** | "1. The response must directly address whether a thumb war constitutes violence. [Hard Rule] 2. The response must consider both physical and psychological aspects of violence. [Hard Rule] 3. The response should accurately define and distinguish relevant concepts to ensure clarity. [Principle] 4. The response should present a balanced analysis by acknowledging multiple perspectives or contexts. [Principle] 5. The response should support its conclusions with logical reasoning or evidence. [Principle] 6. The response should maintain a respectful and empathetic tone. [Principle] " |
| **Judge** | "The response must directly address whether a thumb war constitutes violence ... Response A fails ... not addressing thumb wars at all. Response B satisfies both .... (Choose B)" |

*Table 7.* Comparison of online RL method with different alignment strategies applied to Qwen2.5-7B-Instruct on instruction following and preference alignment benchmarks. Best results are in **bold**.

| Method | IFEval (Prompt) | | IFEval (Inst.) | | IFBench | AlpacaEval | | AVG |
| --- | --- | --- | --- | --- | --- | --- | --- | --- |
| | Loose | Strict | Loose | Strict | | Vanilla | Length | |
| Qwen2.5-7B-Instruct | 75.0 | 72.5 | 81.8 | 79.9 | 28.2 | 33.5 | 36.2 | 46.8 |
| +GRPO (RM-R1) | 76.7 | 73.6 | 83.2 | 80.2 | 30.6 | 53.2 | 42.7 | 52.3 |
| +GRPO (RUBRIC-ARM) | **79.3** | **76.2** | **85.3** | **83.0** | **34.8** | **56.2** | **44.8** | **55.4** |

*Table 8.* Computing speed on 100 samples (vLLM). Results with "⋆" were taken from Liu et al. (2025a).

| | Compute Time (s) |
| --- | --- |
| JudgeLRM-7B⋆ | 25.71 |
| RRM-7B⋆ | 203.40 |
| RM-R1-7B (Qwen-2.5-Inst)⋆ | 260.37 |
| RM-R1-7B (DeepSeek-Dist)⋆ | 170.76 |
| RM-R1-14B (Qwen-2.5-Inst)⋆ | 322.79 |
| RM-R1-14B (DeepSeek-Dist)⋆ | 382.02 |
| RUBRIC-RM-8B | 105.12 |
| RUBRIC-ARM-8B | 33.50 |

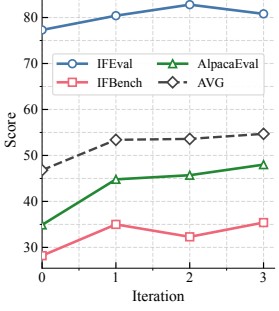

*Figure 5.* Performance of iterative DPO with RUBRIC-ARM across three iterations.

on average. The gains are consistent across instruction-following and human-preference alignment metrics, which indicates that RUBRIC-ARM provides a more effective on-line learning signal for GRPO.

### 6.6. Effect of Iterative Policy Optimization

Fig. 5 evaluates iterative DPO with RUBRIC-ARM over three optimization iterations. Overall, the average performance increases monotonically across iterations, indicating that iteratively refining the policy with RUBRIC-ARM-based supervision yields progressively better alignment. These results suggest that RUBRIC-ARM provides a sufficiently stable signal to support multi-round offline optimization without performance degradation.

### 6.7. Efficiency Comparison

We conclude with an inference-cost analysis and case studies. Table 8 reports wall-clock time on 100 Reward-Bench2 prompts. Despite using two `Qwen-3-8B` modules (rubric generator + judge), RUBRIC-ARM runs in 33.50s, faster than most reasoning-based and rubric-based baselines. While JudgeLRM is slightly faster, it does not provide

the explicit, interpretable rubric-conditioned signals that RUBRIC-ARM is designed for downstream policy optimization. Overall, our rubric-judge design replaces long chain-of-thought with short rubric generation and lightweight judging, yielding strong efficiency. RUBRIC-ARM is also faster than RUBRIC-RM, which typically generates longer rubric lists and incurs higher overhead.

### 6.8. Case Study

We qualitatively analyze failures of baseline reward models on challenging examples. Table 6 shows a RewardBench Chat-Hard instance about *thumb war*: reasoning-based models (e.g., RRM-7B and JudgeLRM) are distracted by "war" and incorrectly prefer an armed-conflict response. In contrast, RUBRIC-ARM generates and enforces a rubric with an explicit hard rule about *thumb war*, leading to the cor-

rect preference. We provide additional IFBench examples in App. D.3, where RUBRIC-ARM reliably extracts hard constraints and judges correctly while RUBRIC-RM fails.

# 7. Conclusion

In this work, we propose RUBRIC-ARM , a novel framework for reward modeling in non-verifiable LLM post-training. Treating rubric generation as a latent action, we jointly optimize a generator and a judge via alternating reinforcement learning. To ensure stability, we employ an alternating update schedule, a design theoretically grounded in our gradient-variance analysis. Empirically, RUBRIC-ARM achieves 4.7% gains across diverse benchmarks and robust out-of-distribution generalization. It also delivers superior supervision for policy alignment in both offline and online RL settings, showing RUBRIC-ARM offers a more reliable reward signal than static approaches.

# Impact Statement

This paper presents work whose goal is to advance the field of Machine Learning. There are many potential societal consequences of our work, none of which we feel must be specifically highlighted here.

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

# A. Details for Group Relative Policy Optimization (GRPO)

GRPO (Shao et al., 2024) is an actor-only policy optimization method that reduces variance by using the *within-prompt* average reward as a baseline. Concretely, for each prompt $q$, GRPO samples a group of responses $\{o_1, o_2, \ldots, o_G\}$ from the old policy $\pi_{\theta_{\text{old}}}(\cdot \mid q)$, computes a group-normalized advantage $\widehat{A}_{i,t}$ for each token, and then performs a PPO-style clipped update. Following Yu et al. (2025a), we upweight informative prompts using a larger clipping threshold $\varepsilon_{\text{high}}$.

$$\mathcal{J}_{\text{GRPO}}(\theta) = \mathbb{E}_{q \sim P(Q), \{o_i\}_{i=1}^G \sim \pi_{\theta_{\text{old}}}(\cdot \mid q)} \left[ \frac{1}{G} \sum_{i=1}^G \frac{1}{|o_i|} \sum_{t=1}^{|o_i|} \min\left( \rho_{i,t}(\theta)\, \widehat{A}_{i,t},\ \text{clip}\left(\rho_{i,t}(\theta), 1-\varepsilon_{\text{low}}, 1+\varepsilon_{\text{high}}\right) \widehat{A}_{i,t} \right) - \beta\, \mathbb{D}_{\text{KL}}[\pi_\theta \| \pi_{\text{ref}}] \right],$$

(21)

where $\rho_{i,t}(\theta) = \frac{\pi_\theta(o_{i,t} \mid q, o_{i,<t})}{\pi_{\theta_{\text{old}}}(o_{i,t} \mid q, o_{i,<t})}$ is the token-level importance ratio.

# B. Detailed Theoretical Derivations

In this section, we provide the complete proofs for the variance analysis presented in Section 5.

## B.1. Preliminaries

Recall the definitions:

- Reward: $R(o) = \mathbb{I}[o = o^\star]$.

- Judge Correctness: $p(r) = \pi_j(o^* \mid c, r)$.

- Generator Score: $u_r(r) = \nabla_{\theta_r} \log \pi_r(r \mid x)$.

- Judge Score: $u_j(o \mid r) = \nabla_{\theta_j} \log \pi_j(o \mid c, r)$.

We utilize the vector form of the **Law of Total Variance**:

**Lemma B.1.** *For random vectors $X$ and $Y$, $\text{Var}(Y) = \mathbb{E}_X[\text{Var}(Y \mid X)] + \text{Var}_X(\mathbb{E}[Y \mid X])$.*

## B.2. Proof of Proposition 5.1 (Strategy A)

*Proof.* In Strategy A, the rubric $\bar{r}$ is fixed. The gradient estimator is $\widehat{g}_A = R(o)u_j(o \mid \bar{r})$, where $o \sim \pi_j(\cdot \mid \bar{r})$. Since $\bar{r}$ is fixed, $u_j(o \mid \bar{r})$ takes two values: $u_j(o^* \mid \bar{r})$ (when correct) and $u_j(\neg o^* \mid \bar{r})$ (when wrong). Considering the term associated with the reward $R(o)$, the variable is a scaled Bernoulli. Conditioned on $\bar{r}$:

- With probability $p(\bar{r})$, $o = o^*$, so $\widehat{g}_A = 1 \cdot u_j(o^* \mid \bar{r})$.

- With probability $1 - p(\bar{r})$, $o \neq o^*$, so $\widehat{g}_A = 0$ (since $R = 0$).

Let $v := u_j(o^* \mid \bar{r})$. The first moment is:

$$\mathbb{E}[\widehat{g}_A \mid \bar{r}] = p(\bar{r})v + (1 - p(\bar{r})) \cdot 0 = p(\bar{r})v.$$

The second moment is:

$$\mathbb{E}[\|\widehat{g}_A\|^2 \mid \bar{r}] = p(\bar{r})\|v\|^2 + (1 - p(\bar{r})) \cdot 0 = p(\bar{r})\|v\|^2.$$

Thus, the variance is:

$$\begin{aligned}
\text{Var}(\widehat{g}_A \mid \bar{r}) &= \mathbb{E}\|\widehat{g}_A\|^2 - \|\mathbb{E}[\widehat{g}_A]\|^2 \\
&= p(\bar{r})\|v\|^2 - \|p(\bar{r})v\|^2 \\
&= (p(\bar{r}) - p(\bar{r})^2)\|v\|^2 \\
&= p(\bar{r})(1 - p(\bar{r}))\|u_j(o^* \mid \bar{r})\|^2.
\end{aligned}$$

$\square$

**B.3. Proof of Proposition 5.2 (Strategy B)**

*Proof.* In Strategy B, we update $\theta_r$. The estimator is $\widehat{g}_B = R(o)u_r(r)$, where $r \sim \pi_r$ and $o \sim \pi_j(\cdot \mid r)$. We apply Lemma B.1 conditioning on $r$.

**Step 1: Conditional Variance (Inner Term).** Conditioned on $r$, $u_r(r)$ is a constant vector. The randomness comes only from $R(o)$.

$$\mathrm{Var}(\widehat{g}_B \mid r) = \mathrm{Var}_{o \mid r}(R(o)u_r(r))$$
$$= \|u_r(r)\|^2 \mathrm{Var}_{o \mid r}(R(o)).$$

Since $R(o) \mid r \sim \mathrm{Bernoulli}(p(r))$, its variance is $p(r)(1 - p(r))$. Thus:

$$\mathrm{Var}(\widehat{g}_B \mid r) = p(r)(1 - p(r))\|u_r(r)\|^2.$$

**Step 2: Conditional Expectation (Outer Term).**

$$\mathbb{E}[\widehat{g}_B \mid r] = \mathbb{E}_{o \mid r}[R(o)]u_r(r) = p(r)u_r(r).$$

**Step 3: Total Variance Decomposition.** By applying the Law of Total Variance (Lemma B.1), we express the total variance as the sum of the expected conditional variance and the variance of the conditional expectation:

$$\mathrm{Var}(\widehat{g}_B) = \mathbb{E}_r[\mathrm{Var}(\widehat{g}_B \mid r)] + \mathrm{Var}_r(\mathbb{E}[\widehat{g}_B \mid r]).$$

Substituting the results derived in Step 1 and Step 2 into the equation above yields the final decomposition:

$$\mathrm{Var}(\widehat{g}_B) = \mathbb{E}_r\Big[p(r)(1 - p(r))\|u_r(r)\|^2\Big] + \mathrm{Var}_r(p(r)u_r(r)).$$

This concludes the proof. $\qquad\square$

**B.4. Proof of Theorem 5.5**

*Proof.* We analyze the sign of the variance difference $\Delta = \mathrm{Var}(\widehat{g}_B) - \mathbb{E}_{\bar{r}}[\mathrm{Var}(\widehat{g}_A \mid \bar{r})]$.

**1. Variance Difference Expansion.** Substituting the expressions from Propositions 5.1 and 5.2:

$$\Delta = \underbrace{\mathbb{E}_r\Big[p(r)(1 - p(r))\|u_r(r)\|^2\Big] + \mathrm{Var}_r(p(r)u_r(r))}_{V_B}$$
$$- \underbrace{\mathbb{E}_r\Big[p(r)(1 - p(r))\|u_j(o^* \mid r)\|^2\Big]}_{V_A}$$
$$= \mathbb{E}_r\Big[p(r)(1 - p(r))\Big(\|u_r(r)\|^2 - \|u_j(o^* \mid r)\|^2\Big)\Big]$$
$$+ \mathrm{Var}_r(p(r)u_r(r)).$$

**2. Incorporating the Exploration Coefficient.** Using the definition of $C_1$ from Assumption 5.3, we substitute $\mathrm{Var}_r(p(r)u_r(r)) = C_1\mathbb{E}_r[p(r)^2\|u_r(r)\|^2]$:

$$\Delta = \mathbb{E}_r\Big[p(r)(1 - p(r))\Big(\|u_r(r)\|^2 - \|u_j(o^* \mid r)\|^2\Big)$$
$$+ C_1 p(r)^2\|u_r(r)\|^2\Big].$$

**3. Verification of Positivity.** To show $\Delta > 0$, we analyze the term inside the expectation (the integrand). We split the

expression into multiple lines to isolate the quadratic components:

$$\begin{aligned}
\text{Integrand} &= (p(r) - p(r)^2)\|u_r(r)\|^2 \\
&\quad - (p(r) - p(r)^2)\|u_j(o^* \mid r)\|^2 \\
&\quad + C_1 p(r)^2 \|u_r(r)\|^2 \\
&= p(r)\Big[(1 - p(r))\|u_r(r)\|^2 \\
&\quad - (1 - p(r))\|u_j(o^* \mid r)\|^2 + C_1 p(r)\|u_r(r)\|^2\Big] \\
&= p(r)\Big[\|u_r(r)\|^2(1 - p(r) + C_1 p(r)) \\
&\quad - \|u_j(o^* \mid r)\|^2(1 - p(r))\Big].
\end{aligned}$$

We now invoke the inequality from Assumption 5.3:

$$\frac{\|u_r(r)\|}{\|u_j(o^* \mid r)\|} > \sqrt{\frac{1 - p(r)}{1 - p(r) + C_1 p(r)}}.$$

Squaring both sides and rearranging:

$$\|u_r(r)\|^2(1 - p(r) + C_1 p(r)) > \|u_j(o^* \mid r)\|^2(1 - p(r)).$$

This implies that the term inside the square brackets is strictly positive. Since $p(r) \in (0, 1)$, the entire integrand is strictly positive for any $r$. Therefore, the expectation is strictly positive:

$$\Delta > 0 \implies \mathrm{Var}(\widehat{g}_B) > \mathbb{E}_{\bar{r}}[\mathrm{Var}(\widehat{g}_A \mid \bar{r})].$$

This concludes the proof. □

# C. Implementation Details

Table 9 and Table 10 show the hyperparameters used in RUBRIC-ARM and policy model training. We implement the GRPO training based on ms-swift[1] library (Zhao et al., 2024) and implement DPO and IterDPO based on LLaMA-Factory[2] (Zheng et al., 2024). We totally conduct 3 iterations for RUBRIC-ARM alternating RL training. Additionally, the sampling parameters used in inference are summarized in Table 11. We used the same sampling parameters as their official implementations and papers for baseline methods.

# D. Additional Experimental Results

## D.1. Performance on WritingPreferenceBench

We present the performance on WritingPreferenceBench in Table 12.

## D.2. Position Bias Analysis

In this section, we study position bias in pairwise judge and reward models, where the predicted preference may depend on the relative order of the two responses (Shi et al., 2025). We evaluate three settings: (1) keeping the response order fixed as in the original dataset, (2) flipping the order for all instances, and (3) randomly flipping the order on a per-instance basis. Table 13 reports results on RewardBench and the IF evaluation benchmarks. Overall, baseline methods exhibit non-trivial position bias. For RRM-7B, changing the order leads to a 46.2-point difference on PPE-IFEval (75.8 vs. 29.6). Likewise, for RM-R1-7B (Qwen-2.5-Inst), flipping the order changes InfoBench by 11.9 points (81.8 vs. 69.9). For RM-R1-7B (DeepSeek-Dist), the order sensitivity remains substantial, with a 9.9-point difference on InfoBench (78.3 vs. 68.4) and a 9.3-point difference on FollowBench (79.0 vs. 69.7). In contrast, our RUBRIC-ARM remains consistently stable across

---

[1] https://github.com/modelscope/ms-swift
[2] https://github.com/hiyouga/LlamaFactory

*Table 9.* Hyper-parameters used in RUBRIC-ARM training.

| Module | Parameter | Value | Module | Parameter | Value |
|---|---|---|---|---|---|
| Rubric Generator | #generations | 6 | Judge | #generations | 7 |
| | Cutoff Length | 512 | | Cutoff Length | 1024 |
| | Batch Size | 288 | | Batch Size | 224 |
| | Optimizer | AdamW | | Optimizer | AdamW |
| | Learning Rate | 1e-6 | | Learning Rate | 1e-6 |
| | Temperature | 1.0 | | Temperature | 1.0 |
| | #iterations | 2 | | #iterations | 2 |
| | Epochs | 1 | | Epochs | 1 |
| | $\epsilon_{high}$ | 0.28 | | $\epsilon_{high}$ | 0.28 |
| | $\epsilon_{low}$ | 0.2 | | $\epsilon_{low}$ | 0.2 |
| | $\beta$ | 0.001 | | $\beta$ | 0.001 |

*Table 10.* Hyper-parameters used in policy model training.

| Method | Parameter | Value | Method | Parameter | Value |
|---|---|---|---|---|---|
| DPO | Cutoff Length | 2048 | GRPO | #generations | 6 |
| | Batch Size | 64 | | Cutoff Length | 2048 |
| | Optimizer | AdamW | | Batch Size | 288 |
| | Learning Rate | 8e-7 | | Optimizer | AdamW |
| | Epochs | 1 | | Learning Rate | 5e-7 |
| | beta | 0.1 | | Temperature | 1.0 |
| | SFT mixing weight | 0.2 | | #iterations | 2 |
| | / | / | | Epochs | 1 |
| | / | / | | $\epsilon_{high}$ | 0.28 |
| | / | / | | $\epsilon_{low}$ | 0.2 |
| | / | / | | $\beta$ | 0.001 |

*Table 11.* Sampling parameters used in RUBRIC-ARM inference.

| Module | Parameter | Value | Module | Parameter | Value |
|---|---|---|---|---|---|
| Rubric Generator | Maximum Tokens | 1024 | Judge | Maximum Tokens | 4096 |
| | Temperature | 0.0 | | Temperature | 1.0 |
| | Top-P | / | | Top-P | 1.0 |
| | Top-K | / | | Top-K | -1 |
| | Enable-thinking | False | | Enable-thinking | False |

different orderings, suggesting substantially reduced position bias and more robust evaluation. This design choice is aligned with our RL training design, where we randomize the response order when collecting reward signals, which further mitigates position bias in downstream policy optimization.

### D.3. Additional Case Study

In this section we compare RUBRIC-ARM with RUBRIC-RM, another rubric-based RM trained with SFT, on a randomly chosen example from IFBench. The case specifies keywords and paragraph length. Results are shown in Table 14. In this IFBench example, which requires specific keywords and exactly two paragraphs, the baseline RUBRIC-RM suffers from a judging hallucination, incorrectly claiming that a valid response is split into three paragraphs. RUBRIC-ARM, on the contrary, accurately extracts these hard constraints and identifies the missing *open-source* keyword in the negative sample, while correctly verifying the structure of the positive one.

*Table 12.* Comparison of different judge and reward models on WritingPreferenceBench. Best results are highlighted in **bold**.

|  | Func. | Promo. | Non-Fic. | Fiction | Funny | Poetry | Script | Role | AVG |
|---|---|---|---|---|---|---|---|---|---|
| *LLM as Judge (black-box model)* | | | | | | | | | |
| Claude-4-Opus-thinking | 65.7 | 64.3 | 64.1 | 60.1 | 54.2 | 64.0 | 43.5 | 51.7 | 61.0 |
| OpenAI-o4-mini | 58.3 | 58.6 | 60.9 | 55.5 | 53.2 | 68.0 | 30.4 | 55.2 | 56.6 |
| Gemini-2.5-Flash | 59.1 | 57.7 | 62.5 | 59.8 | 52.2 | 56.0 | 34.8 | 51.7 | 57.5 |
| *White-box Reward Models* | | | | | | | | | |
| Skywork-Llama-3.1-8B | 53.6 | 56.3 | 60.6 | 49.0 | 52.2 | 56.0 | **65.2** | 41.4 | 53.1 |
| Skywork-Gemma-2-27B | 49.0 | 53.9 | 59.6 | 33.9 | 55.1 | 36.0 | 21.7 | 51.7 | 46.8 |
| RM-R1-DeepSeek-Qwen-7B | 62.5 | 55.1 | 59.2 | 55.4 | 58.0 | 56.0 | **65.2** | 41.4 | 57.4 |
| RM-R1-Qwen2.5-7B | 67.0 | 57.2 | 53.9 | 60.0 | 54.6 | 72.0 | 47.8 | **65.5** | 59.8 |
| RRM-7B | 50.0 | 35.3 | 50.0 | 49.5 | 38.5 | 36.4 | 45.5 | 53.8 | 44.7 |
| *Rubric-based Models* | | | | | | | | | |
| Rubric-RM | 58.3 | 58.5 | 57.9 | 58.3 | 58.0 | 76.0 | 47.8 | 55.2 | 60.3 |
| Rubric-ARM | **67.8** | **63.1** | **65.8** | **60.9** | **61.0** | **80.0** | 47.8 | 55.2 | **63.2** |

*Table 13.* Position bias analysis for different judge and reward models. RUBRIC-ARM shows much lower sensitivity to the ordering of response pairs.

| | RewardBench | | IF Evaluation Benchmarks | | | | Avg. Variation |
|---|---|---|---|---|---|---|---|
| | Chat | Chat Hard | FollowBench | PPE-IFEval | InfoBench | IFBench | |
| *White-box Judge/Reward LLM: RRM-7B* | | | | | | | |
| Mixed Ord | 77.7 | 69.5 | 65.5 | 51.0 | 68.2 | 53.2 | |
| Fixed Ord-1 | 73.9 | 61.6 | 53.8 | 29.6 | 62.3 | 30.2 | |
| Fixed Ord-2 | 82.1 | 72.1 | 64.7 | 75.8 | 74.2 | 74.2 | |
| Variation | 8.2 | 10.5 | 11.7 | 46.2 | 11.9 | 44.0 | 22.08 |
| *White-box Judge/Reward LLM: RM-R1-7B (Qwen-2.5-Inst)* | | | | | | | |
| Mixed Ord | 83.0 | 70.0 | 56.3 | 55.2 | 71.3 | 55.2 | |
| Fixed Ord-1 | 82.1 | 63.4 | 57.1 | 54.8 | 81.8 | 53.8 | |
| Fixed Ord-2 | 82.4 | 71.1 | 56.3 | 50.4 | 69.9 | 54.1 | |
| Variation | 0.9 | 7.7 | 0.8 | 4.8 | 11.9 | 1.4 | 4.58 |
| *White-box Judge/Reward LLM: RM-R1-7B (DeepSeek-Dist)* | | | | | | | |
| Mixed Ord | 85.3 | 67.3 | 69.7 | 51.0 | 70.3 | 56.5 | |
| Fixed Ord-1 | 87.1 | 67.3 | 79.0 | 52.8 | 78.3 | 53.2 | |
| Fixed Ord-2 | 82.7 | 69.5 | 70.6 | 54.7 | 68.4 | 60.6 | |
| Variation | 4.4 | 2.2 | 9.3 | 3.7 | 9.9 | 7.4 | 6.15 |
| *Rubric-based Method:* RUBRIC-RM | | | | | | | |
| Mixed Ord | 88.2 | 74.1 | 76.1 | 67.0 | 80.8 | 65.4 | |
| Fixed Ord-1 | 87.4 | 74.6 | 79.8 | 70.8 | 80.9 | 66.4 | |
| Fixed Ord-2 | 88.7 | 73.5 | 75.6 | 67.2 | 78.5 | 64.4 | |
| Variation | 1.3 | 1.1 | 4.2 | 3.8 | 2.4 | 2.0 | 2.47 |
| *Rubric-based Method:* RUBRIC-ARM *(Ours)* | | | | | | | |
| Mixed Ord | 89.4 | 79.6 | 85.7 | 70.8 | 86.1 | 65.9 | |
| Fixed Ord-1 | 89.9 | 79.4 | 84.9 | 71.8 | 86.1 | 65.3 | |
| Fixed Ord-2 | 88.4 | 80.3 | 85.7 | 71.0 | 87.9 | 66.9 | |
| Variation | 1.5 | 0.9 | 0.8 | 1.0 | 1.8 | 1.6 | 1.27 |

*Table 14.* Case study with error highlighting. In the baseline RUBRIC-RM, the rubric output confused the judge and led to an incorrect prediction.

| Case Study on IFBench | |
| --- | --- |
| **Prompt** | What is Nextcloud, why should I use it, and how does it compare to other cloud storage solutions? Please provide the response with a tone of excitement and enthusiasm. Your response should contain *exactly 2 paragraphs* and include the keywords 'cloud storage' and 'open-source'. |
| **Resp A (snippet)** | "Nextcloud is an incredibly exciting and innovative cloud storage solution . . . As an open-source platform . . . Imagine having your very own private cloud . . ." (*2 paragraphs*) |
| **Resp B (snippet)** | "Nextcloud is an incredibly powerful and flexible solution . . . While services like Google Drive and Dropbox are great . . ." (*2 paragraphs; missing "open-source"*) |
| **Label** | Resp A. |
| **RUBRIC-RM** | |
| **Rubric** | 1. The response must define what Nextcloud is. [Hard Rule] 2. The response must explain why the user should use Nextcloud. [Hard Rule] 3. The response must compare Nextcloud to other cloud storage solutions. [Hard Rule] 4. The response must use a tone of excitement and enthusiasm. [Hard Rule] 5. The response must consist of exactly two paragraphs. [Hard Rule] 6. The response must include the keywords 'cloud storage' and 'open-source'. [Hard Rule] 7. The response should present information in a clear, logical structure that guides the reader through definition, benefits, and comparison. [Principle] 8. The response should use vivid, engaging language to maintain reader interest. [Principle] 9. The response should employ precise and varied vocabulary to convey ideas effectively. [Principle] 10. The response should provide specific, relevant details to substantiate its claims. [Principle] 11. The response should maintain coherence and flow, ensuring each paragraph builds on the previous one. [Principle] 12. The response should balance conciseness with sufficient elaboration to cover all required points. [Principle] |
| **Judge** | ". . . Response A is split into three paragraphs, not exactly two . . . Response A fails the gatekeeper criterion (paragraph count) . . . (Choose B) |
| **RUBRIC-ARM** | |
| **Rubric** | 1. The response must address what Nextcloud is, why it should be used, and how it compares to other cloud storage solutions. [Hard Rule] 2. The response must maintain an enthusiastic and excited tone throughout. [Hard Rule] 3. The response must consist of exactly two paragraphs. [Hard Rule] 4. The response must include the keywords 'cloud storage' and 'open-source'. [Hard Rule] 5. The response should present information in a clear and logical structure, grouping related ideas coherently. [Principle] 6. The response should use vivid and engaging language to sustain reader interest. [Principle] 7. The response should provide relevant comparisons or contrasts to contextualize the subject. [Principle] 8. The response should maintain conciseness and avoid unnecessary repetition. [Principle] |
| **Judge** | ". . . Response B mentions 'cloud' and 'open' but not the exact keyword 'open-source,' so fails. . . . Response A fulfills all hard rules, including the precise keywords . . . (Choose A)" |

