# OpenReview forum: "Alternating Reinforcement Learning for Rubric-Based Reward Modeling in Non-Verifiable LLM Post-Training"
_ICML.cc/2026/Conference — ICML 2026 regular_

### Official Review · Reviewer_uKKq · 2026-03-11

**Soundness:** 3
**Presentation:** 3
**Significance:** 2
**Originality:** 3
**Overall Recommendation:** 4
**Confidence:** 4

**Summary:**

This paper proposes the R-ARM framework to address the challenge of quantifying response quality in non-verifiable domains. It models rubric generation as latent actions to enable multi-dimensional quality assessment. Through joint optimization and an alternating learning strategy between the rubric generator and the judge, the framework achieves mutual enhancement and stable training, ultimately improving evaluation and generation quality for reward modeling in LLM post-training.

**Compliance With Llm Reviewing Policy:**

Affirmed.

**Key Questions For Authors:**

1.How is the warm-up stage implemented for the generator and the judge?
2. Since both the generator and the judge are based on Qwen-3-8B, is the information passed between them during alternating optimization in natural language form or human-readable structured information?
3. Why does swapping the optimization order lead to a decrease in mean performance? Is there any supporting theoretical assumption?
4. Besides unstructured data such as images that can be understood by large models, do non-verifiable domains also include semi-structured data such as code snippets for specific tasks? For such data, what criteria does R-ARM use to design the reward function?
5.How do you reconcile the focus on "Non-Verifiable" domains with the heavy reliance on constraint-following benchmarks (IFEval) where correctness is objectively verifiable?
6. In alternating RL, it is common for the generator to find a narrow set of "shortcut" representations that the judge easily scores highly on. Did you observe mode collapse in the generated rubrics (e.g., generating identical, generic criteria for different prompts), and how does the KL-divergence penalty in GRPO specifically mitigate this across diverse tasks?

**Limitations:**

This work mainly focuses on reward modeling in non-verifiable domains. Although such domains can be further subdivided, experimental validation and reward mechanism design for various sub-types have not been sufficiently covered, which constitutes a limitation of this work.

**Strengths And Weaknesses:**

Strength
To address the issue that scalar scores from standard reward models fail to capture the multi-dimensional quality of open-domain tasks, this work provides a novel perspective that models rubric generation as latent actions, breaking through the limitations of static rubrics.
It proposes an optimization method for LLM post-training that effectively alleviates non-stationarity in simultaneous training via an alternating optimization strategy, with theoretical proof of reduced gradient variance.

Weakness
The core idea—alternating optimization of two interdependent models—is heavily borrowed from classic ML paradigms (e.g., Expectation-Maximization, GANs). Applying this via GRPO to a Generator-Judge pipeline is an engineering synthesis rather than a fundamental algorithmic breakthrough.
The title explicitly targets "Non-Verifiable LLM Post-Training". Yet, a substantial portion of the evaluation relies on benchmarks like IFEval and InfoBench , which are specifically designed to test verifiable constraints (e.g., "exactly 2 paragraphs", "include specific keywords"). This contradicts the paper's core motivation.

---

> ### Author Rebuttal · Authors · 2026-03-31
>
> We highly appreciate your effort and time spent reviewing our paper and thank you for your expertise and constructive comments. In the following, we address your comments and questions one by one.
>
> >W1: The core idea (alternating optimization of two interdependent models) is borrowed from classic ML …
>
> We agree that alternating optimization itself is not new, and we do not claim it as a novel optimization principle. Our novelty is in being, to our knowledge, the first to study **rubric-based reward modeling as a joint optimization**, where **rubric generation and judge training are optimized in a unified framework** rather than treated as separate stages or fixed components. The technical novelty lies in the RL formulation for this discrete, non-stationary generator-judge setting: (1) phase-specific GRPO updates, (2) rubric caching, (3) a judge-first schedule, and (4) theoretical analysis. These are dedicated designs for joint rubric generation and judge learning, supported by both theory and experiments: the variance analysis motivates the schedule, reversing the update order hurts performance (74.8 → 72.4), and the full method improves over RUBRIC-RM (70.1 → 74.8).
>
> >W2 & Q5: The title explicitly targets "Non-Verifiable LLM Post-Training". Yet, a substantial portion of the evaluation relies on benchmarks like IFEval and InfoBench designed to test verifiable constraints…
>
> Our focus on **non-verifiable post-training** is to distinguish from math/code domains where **verifiable rewards** are available. Consistent with this scope, both the rubric generator and judge are optimized only from **pairwise preference data**, without executable tests or correctness oracles.
>
> We include IFEval and InfoBench for comprehensive evaluation—not because our method relies on verifiable signals. Notably, **IFEval is the only fully verifiable benchmark, while InfoBench includes non-verifiable constraints**, making it partially aligned with our setting. Strong performance here should be viewed as a *strength*.
>
> The majority of evaluation targets non-verifiable tasks (e.g., chat, writing), where RUBRIC-ARM improves average RM scores (70.1 → 74.8) and achieves the best OOD results on WritingPreferenceBench, with case studies showing rubrics capture prompt-specific, non-deterministic criteria.
>
> >Q1 & Q2: How is the warm-up stage implemented for the generator and the judge? Since both the generator and the judge are based on Qwen-3-8B, is the information passed between them during alternating optimization in natural language form or human-readable structured information?
>
> The generator and judge are **initialized separately via SFT** on the OpenRubrics dataset. The information passed between them is **explicit rubric text (human-readable structured criteria)**, not hidden states or latent vectors.
>
> >Q3: Why does swapping the optimization order lead to a decrease in mean performance? Is there any supporting theoretical assumption?
>
> Sec. 5 shows that updating the generator first creates **higher-variance training signals**, because the still-weak judge produces noisy rewards on new rubrics. Training the judge first with fixed rubrics reduces this noise before optimizing the generator. This is formalized by Assumption 5.3, Theorem 5.5, and Remark 5.6. The ablation in Tab 2 confirms that swapping the order lowers average performance.
>
> >Q4: … do non-verifiable domains also include semi-structured data such as code snippets for specific tasks? For such data, what criteria does Rubric-ARM use to design the reward function?
>
> We clarify that image inputs are not studied in this paper; our setting is text-based. Code snippets need not be purely verifiable: when execution-based tests are unavailable or insufficient, rubrics can define criteria such as correctness, spec adherence, codebase consistency, and safety, and the judge scores solutions against them.
>
> >Q6: ... Did you observe mode collapse in the generated rubrics, and how does the KL-divergence penalty in GRPO specifically mitigate this?
>
> We did not observe mode collapse. Rubrics remain **prompt-specific** rather than converging to a generic template. Table 7 shows a concrete example: for the *thumb war* prompt, RUBRIC-ARM generates a hard rule requiring the response to address whether a thumb war is violent—not a generic reusable criterion. Quantitatively, we sampled prompts at early and final checkpoints. Criteria count remains variable (**8.52±1.97** and **8.29±2.33**), and within-rubric self-BLEU (lower means more diverse) remains low (**0.191** and **0.206**), indicating no collapse into repetitive criteria. The **KL term in GRPO** keeps both models close to warm-start policies (**$\beta=1e-3$**), discouraging drift toward shortcut solutions. Combined with alternating updates, this prevents the generator from exploiting a transient weak judge.

---

> > ### Author Rebuttal · Reviewer_uKKq · 2026-04-04
> >
> > Thank you for the detailed rebuttal, which successfully addresses my primary concerns regarding the 'non-verifiable' task definition and potential mode collapse. The new quantitative data, particularly the self-BLEU scores, convincingly strengthens your empirical claims.

---

### Official Review · Reviewer_BGEm · 2026-03-12

**Soundness:** 3
**Presentation:** 3
**Significance:** 3
**Originality:** 3
**Overall Recommendation:** 4
**Confidence:** 3

**Summary:**

This paper introduces the method called "RUBRIC-ARM" which improves reward models by jointly learning a rubric generator and a judge using alternating RL. This method learns task-specific rubrics to guide judgements. Experiments on instruction following and writing benchmarks show improvements.

**Compliance With Llm Reviewing Policy:**

Affirmed.

**Final Justification:**

Please see my reply to rebuttal.

**Key Questions For Authors:**

See weaknesses above.

**Limitations:**

Yes

**Strengths And Weaknesses:**

Strengths:
1. Clear idea which is well-motivated. The adaptive reward rubrics help mitigate the limit of traditional rewarding.
2. Strong empirical results across multiple benchmark datasets.
3. Improvement in downstream models after RL supports the method.

Weaknesses:
1. The alternating RL with two models make the training a lot more expensive than standard reward modeling.
2. Limited theoretical analysis for convergence guarantee.
3. Baselines could be expended to include more resent reasoning-heavy judges.
4. How does it scale to even larger models and much larger datasets?

---

> ### Author Rebuttal · Authors · 2026-03-31
>
> We highly appreciate your effort and time spent reviewing our paper and thank you for your expertise and constructive comments. In the following, we address your comments and questions one by one.
>
> > W1: The alternating RL with two models is more expensive…
>
> We appreciate the concern. First, we agree that alternating RL introduces additional training cost relative to single-model reward modeling, and we will clarify this more explicitly. However, the overhead is moderate in practice: **one iteration** of our method  finishes in **\~10-12 hours** on 8xA100 80GB and already delivers strong performance gains, showing that many rounds are not required.
>
> | |RB Chat|RB Chat Hard|FollowBench|PPE-IFEval|InfoBench|IFBench|RM-Bench Chat|RB2 Precise IF|RB2 Focus|HelpSteer3|AVG|
> |:-:|:-:|:-:|:-:|:-:|:-:|:-:|:-:|:-:|:-:|:-:|:-:|
> |Rubric-RM 8B|88.2|74.1|76.1|67.0|80.8|65.4|65.7|34.4|82.2|67.0|70.1|
> |Rubric-ARM 8B (Iter1)|89.8|78.7|87.1|69.2|86.1|64.3|69.5|25.6|84.8|70.88|72.6 (+2.5)|
> |Rubric-ARM 8B|89.4|79.6|85.7|70.8|86.1|65.9|69.2|41.9|89.4|69.8|74.8 (+4.7)|
>
>
> More importantly, the main purpose of a reward model is to **serve downstream LLM post-training / RL**, where **inference cost is often more critical than one-time reward-model training cost**. In this respect, our method remains practical: despite using two components, **RUBRIC-ARM is substantially faster at inference than RUBRIC-RM and several reasoning-based reward models** (e.g., 33.5s vs. 105.1s on 100 samples in Sec. 6.7), so it does not add much inference overhead.
>
>
>
>
> > W2: Limited theoretical analysis for convergence guarantee.
>
> Thank you for raising this point. Under additional structural assumptions, specifically, when the loss function is convex and satisfies standard regularity conditions, we can invoke classical results on block coordinate descent (e.g., On the Convergence of Block Coordinate Descent Type Methods) to establish convergence guarantees. In this regime, our method can be shown to achieve faster convergence than naive alternatives due to its more efficient update structure.
>
> However, in the general non-convex setting considered in our work, providing global convergence guarantees is fundamentally challenging and remains an open problem even for closely related alternating optimization schemes. Our theoretical goal is mainly to justify the judge-first schedule through a variance-reduction argument, which we support both analytically (Sec. 5) and empirically (Table 2). We will clarify this scope in the final version.
>
>
> > W3: Baselines could be expanded to include more recent reasoning-heavy judges.
>
> We'd like to mention our baselines, such as RRM (NeurIPS’ 25) and RM-R1 (ICLR’ 26), are among the most relevant, strong and recent reasoning-based reward models.
> Per the reviewer’s suggestion, we further add AgentRM (ACL’ 25), Think-RM (NeurIPS’ 25), and J1 (ICLR’ 26). Note that different models use different evaluation sets, and we report benchmarks overlapping including ChatHard, PPE-IF and Followbench.
>
> As shown in Table below, on the three overlapping benchmarks, Rubric-ARM outperforms similarly sized models on average by 6.3–14.5%, demonstrating its effectiveness.
>
> | |ChatHard|PPE-IF|Followbench|Avg|
> |---|---:|---:|---:|---:|
> |J1-8B|80.3|54.0|58.3|64.2|
> |Think-RM-8B|76.1|63.9|70.4|70.1|
> |AgentRM-8B|74.3|73.0|69.8|72.4|
> |Rubric-ARM-8B|79.6|70.8|85.7|78.7|
>
> [1] Agentic reward modeling: Integrating human preferences with verifiable correctness signals for reliable reward systems. ACL’25
>
> [2] Think-rm: Enabling long-horizon reasoning in generative reward models. NeurIPS’25
>
> [3] J1: Incentivizing Thinking in LLM-as-a-Judge via Reinforcement Learning. ICLR’26
>
> > W4: How does it scale to larger models and larger data?
>
> Following the reviewer’s suggestion, we added results with **Qwen-3-4B** due to budget limits. The gains **remain consistent from 4B to 8B**, suggesting that RUBRIC-ARM is **not tied to a single model scale** and continues to improve performance as the backbone becomes stronger. The 8B version further improves from 71.6 to **74.8**, indicating good scaling with model size.
> For data scaling, our results are also encouraging: the **first alternating round uses only 1/3 of the training data**, yet already yields strong gains, and additional rounds continue to improve performance. This suggests that the method is **data-efficient early on** while still benefiting from more data and training, which is exactly the scaling behavior we would hope to see. We will include these additional results in the final version.
>
> | |RB Chat|RB Chat Hard|FollowBench|PPE-IFEval|InfoBench|IFBench|RM-Bench Chat|RB2 Precise IF|RB2 Focus|HelpSteer3|AVG|
> |:-:|:-:|:-:|:-:|:-:|:-:|:-:|:-:|:-:|:-:|:-:|:-:|
> |4B Iter 1|86.7|73.0|86.1|66.5|83.6|62.6|66.2|33.8|86.1|68.0|71.3|
> |4B|88.4|73.7|82.2|69.5|83.2|65.6|68.1|33.1|82.0|70.3|71.6|
> |8B Iter1|89.8|78.7|87.1|69.2|86.1|64.3|69.5|25.6|84.8|70.8|72.6|
> |8B|89.4|79.6|85.7|70.8|86.1|65.9|69.2|41.9|89.4|69.8|74.8|

---

> > ### Author Rebuttal · Reviewer_BGEm · 2026-04-03
> >
> > The rebuttal adequately addresses most of main concerns. The added experimental results (including stronger baselines and scaling analysis) significantly strengthen the empirical support, and the clarification on efficiency and training cost is convincing. While theoretical guarantees remain limited, this is reasonable for the setting and does not undermine the overall contribution. I will raise my score accordingly.

---

### Official Review · Reviewer_ZdfR · 2026-03-13

**Soundness:** 3
**Presentation:** 3
**Significance:** 2
**Originality:** 2
**Overall Recommendation:** 4
**Confidence:** 3

**Summary:**

This paper studies rubric-based reward modeling for non-verifiable LLM post-training domains, where scalar rewards or direct pairwise judgments may miss multiple aspects of response quality. The proposed method, RUBRIC-ARM, jointly trains a rubric generator and a judge, treats rubric generation as a latent action, and uses alternating RL with a judge-first update order to reduce instability from jointly updating both parts. The paper also uses the learned system as a reward source for downstream offline and online policy optimization, and reports gains over prior rubric-based and reasoning-based reward models on reward-model benchmarks as well as policy-alignment benchmarks.

**Compliance With Llm Reviewing Policy:**

Affirmed.

**Final Justification:**

The paper studies training the rubric generator and judge together with alternating RL, and the experiments show the method is useful in practice across reward-model and policy-training settings. My main concerns were the missing direct comparison to joint updates, the limited empirical support for the judge first schedule, and whether the gains would extend beyond one backbone. The rebuttal addressed these concerns well by adding the joint optimization comparison, providing more support for the variance argument, showing results on a 4B model, and clarifying the extra training cost. That increased my confidence in the paper, but it mostly reinforced rather than changed my overall view, so I am keeping my original score.

**Key Questions For Authors:**

1. Could you add a direct comparison to simultaneous joint RL updates of the rubric generator and judge?
2. Since the method is trained to fit dataset preferences, do you have any analysis of label noise or inconsistent preferences in OPENRUBRICS? Relatedly, how robust is RUBRIC-ARM when the supervision becomes noisier?
3. What is the training cost of RUBRIC-ARM relative to RUBRIC-RM? Table 8 reports inference cost, but the training overhead of alternating RL versus the SFT-only RUBRIC-RM baseline is not discussed, which is important for practical adoption.

**Limitations:**

The current impact statement is minimal. The paper may discuss possible misuse and failure modes. For example, reward models trained in non-verifiable domains may encode biases, which can then be reinforced during policy optimization. This creates risks of systematically favoring certain styles and values while suppressing others, especially in open-ended settings where there is no clear ground truth.

**Strengths And Weaknesses:**

**Strengths:**

1. The core idea of jointly optimizing rubric generation and judging via RL is well-motivated. The theoretical variance analysis provides formal justification for the judge-first ordering, and the method is easy to follow.

2. The experiments cover a wide range of evaluations including 9 reward model benchmarks, 6 policy benchmarks, both offline and online RL settings, plus ablations, efficiency analysis, and case studies. The comparisons against strong baselines are thorough.

**Weakness:**

1. The experiments do not include the most direct comparison to a non-alternating joint RL update, this makes the link between the theoretical motivation and the empirical results a bit less convincing.
2. The theoretical argument provides an intuitive variance-based motivation for the judge-first schedule, but the support remains limited because Assumption 5.3 is not empirically validated, and the justification in Remark 5.4 for comparable gradient norms is informal and not fully convincing, especially given the asymmetry in conditioning context and generation roles between the judge and rubric generator.
3. All experiments use Qwen-3-8B as the base model. It is unclear whether the alternating RL approach would yield similar gains on other model families or scales, limiting the generalizability of the findings.

---

> ### Author Rebuttal · Authors · 2026-03-31
>
> We highly appreciate your effort and time spent reviewing our paper and thank you for your expertise and constructive comments. In the following, we address your comments and questions one by one.
> > W1&Q1: direct comparison to simultaneous joint RL updates of the rubric generator and judge
>
> We include results in the following table, where alternating training outperforms joint optimization by +8.1% on average.
>
> | |RB Chat|RB Chat Hard|FollowBench|PPE-IFEval|InfoBench|IFBench|RM-Bench Chat|RB2 Precise IF|RB2 Focus|HelpSteer3|AVG|
> |:-:|:-:|:-:|:-:|:-:|:-:|:-:|:-:|:-:|:-:|:-:|:-:|
> |Rubric-ARM 8B (Alternating)|89.4|79.6|85.7|70.8|86.1|65.9|69.2|41.9|89.4|69.8|74.8|
> |Joint Optim. 8B|84.6|68.6|73.1|64.8|76.4|66.9|62.4|26.9|77.2|66.1|66.7|
>
> We avoid fully joint updates for both algorithmic and practical reasons. Early in training, the judge is weak, and its rapidly changing outputs introduce high noise and variance in rubric learning. Alternating updates mitigate this instability. Additionally, joint training would require frequent recomputation of rubric-conditioned rollouts and reward labels as both parts evolve, adding significant overhead. The alternating scheme (Sec. 4.2) is thus more stable and efficient.
>
> > W2: The theoretical argument provides an intuitive variance-based motivation for the judge-first schedule, but the support remains limited…
>
> We clarify that the condition in Assumption 5.3 is intentionally designed to be a mild requirement. Specifically, because the RHS of the inequality is strictly less than 1. In practice, the Rubric Generator ($u_r$) typically exhibits a larger gradient norm than the Judge ($u_j$) because it performs a more generative, high-entropy task (creating multi-dimensional criteria) compared to the Judge's more constrained evaluation task. Empirically, we find that the average gradient norm is 0.28 for rubric generation and 0.18 for the judge. These measurements are consistent with Assumption 5.3 on average during early training, thereby providing stronger theoretical support for the observed stability of the judge-first schedule.
>
>
> > W3: Other backbones
>
> Following the reviewer's suggestion, and due to budget constraints, we add experimental results with Qwen-3-4B. Results below demonstrate that the 4B model also benefits from the alternating RL, confirming the generalizability of our approach.
>
> | |RB Chat|RB Chat Hard|FollowBench|PPE-IFEval|InfoBench|IFBench|RM-Bench Chat|RB2 Precise IF|RB2 Focus|HelpSteer3|AVG|
> |:-:|:-:|:-:|:-:|:-:|:-:|:-:|:-:|:-:|:-:|:-:|:-:|
> |Rubric-RM 4B|87.2|68.9|78.0|64.8|79.3|63.2|62.6|35.6|79.6|64.7|68.4|
> |Rubric-RM vote@5 |88.7|70.2|80.7|66.2|83|64.9|63.8|38.1|82.6|65|70.3|
> |Rubric-ARM 4B|88.4|73.7|82.2|69.5|83.2|65.6|68.1|33.1|82|70.3|71.6 (+3.2)|
> |Rubric-ARM vote@5|89.9|74.3|86.6|71.6|85|67.6|68.5|40.6|84.4|71.6|74.0 (+3.7)|
>
>
> > Q2: Label noise or inconsistent preferences in OPENRUBRICS? How robust is RUBRIC-ARM when the supervision becomes noisier?
>
> In OpenRubrics, labels are either human-annotated or pre-filtered via multiple LLM/RM ensembles, resulting in relatively low noise. For robustness, when supervision becomes noisier, RUBRIC-ARM can monitor the agreement between model rollouts and preference labels, and discard samples where all rollouts disagree with the label in the current iteration, reducing the impact of noisy supervision.
>
>
>
> > Q3: The training overhead of alternating RL is not discussed, which is important for practical adoption.
>
>
> Compared with the SFT-only RUBRIC-RM baseline, the main extra cost of RUBRIC-ARM comes from the RL stage. In practice, however, this overhead is moderate: on 8xA100-80GB, the SFT stage takes about **6–7 hours**, while **one alternating RL round** for the two models takes about **10–12 hours**. Importantly, **only one single RL iteration already improves the average score from 70.1 to 72.6 (+2.5) over RUBRIC-RM, showing that most of the gain appears early**.
> Furthermore, for practical adoption, **inference cost is typically more critical than one-time reward-model training cost**, since the reward model is used repeatedly during downstream LLM post-training. On this axis, RUBRIC-ARM remains efficient: despite the additional RL training, Sec. 6.7 shows **RUBRIC-ARM is substantially faster at inference than RUBRIC-RM and several reasoning-based reward models** (e.g., 33.5s vs. 105.1s on 100 samples). We will add this discussion and the one-iteration result in the final version.
>
>
>
> | |RB Chat|RB Chat Hard|FollowBench|PPE-IFEval|InfoBench|IFBench|RM-Bench Chat|RB2 Precise IF|RB2 Focus|HelpSteer3|AVG|
> |:-:|:-:|:-:|:-:|:-:|:-:|:-:|:-:|:-:|:-:|:-:|:-:|
> |Rubric-RM 8B|88.2|74.1|76.1|67.0|80.8|65.4|65.7|34.4|82.2|67.0|70.1|
> |Rubric-ARM 8B (Iter1)|89.8|78.7|87.1|69.2|86.1|64.3|69.5|25.6|84.8|70.8|72.6 (+2.5)|
> |Rubric-ARM 8B|89.4|79.6|85.7|70.8|86.1|65.9|69.2|41.9|89.4|69.8|74.8 (+4.7)|

---

> > ### Author Rebuttal · Reviewer_ZdfR · 2026-04-02
> >
> > Thanks for the rebuttal, I’ll keep my positive score.

---

### Decision · Program_Chairs · 2026-04-30

**Decision:**

Accept (regular)

**Comment:**

The paper proposes a well-motivated framework for rubric-based reward modeling in non-verifiable LLM post-training, jointly optimizing a rubric generator and judge through alternating RL. Reviewers found the idea motivating and practically meaningful, and the empirical results are strong across reward-model and downstream policy benchmarks. There were some concerns raised by the reviewers, including the missing baseline comparison (such as non-alternating joint RL update, as pointed out by Reviewer ZdfR and Reviewer uKKq), the training cost (Reviewer ZdfR and Reviewer BGEm), and the generalizability to other model families or larger models (Reviewer ZdfR and Reviewer BGEm). During rebuttal, these concerns were addressed well by the authors through additional experiments and clarifications. Therefore, I recommend acceptance.